# Enhancing Home Education in Italian Context: Teachers' Perception of a Hybrid Inclusive Classroom

Vincenza Benigno *[ID], Giovanni Paolo Caruso, Francesca Maria Dagnino [ID], Edoardo Dalla Mutta and Chiara Fante

Istituto per le Tecnologie Didattiche, CNR, Via De Marini 6, 16149 Genova, Italy
* Correspondence: benigno@itd.cnr.it

**Abstract:** Homebound Education in Italy is based on one-to-one teaching conducted by teachers who visit the sick student at home. This modality does not include interactions between homebound students and classmates, thus inhibiting the educational aspect of peer relationships. With the outbreak of the COVID-19 pandemic and the consequent dispositions of the Ministry of Education regarding remote education and integrated digital didactic (DID), new opportunities became available for homebound students. In this research, we applied and tested in the context of homebound education a model of hybrid inclusive classroom developed in a previous project ((TRIS), addressing homebound students that could not permanently attend school and followed lessons remotely. The present study involved two homebound students affected by chronic and acute diseases. During the 2020/21 school year, the model was proposed to the two school councils (22 teachers in all) and the trial monitored; at the end of the school year, semi-structured interviews were conducted with teachers and transcriptions analyzed using a deductive approach based on the model. Results confirm some findings of the TRIS project, while new aspects emerged linked to the specific context. Overall, the implemented hybrid classroom seems to have positively affected both the learnings and students' inclusion.

**Keywords:** home education; ICTs; hybrid inclusive classroom; chronic illness; COVID-19; teachers



## 1. Introduction

The spread of the COVID-19 virus has produced severe restrictions connected with the need to maintain social distancing: nationwide closures have impacted almost 70% of the world's student population [1]. The impossibility to continue traditional classroom teaching has made it necessary to adopt distance teaching scenarios, called "emergency remote education" by some [2], in order to guarantee learning continuity for students.

The networks of the OECD (Organization for Economic Co-operation and Development) and the Global Education Innovation Initiative at the Harvard Graduate School of Education conducted an assessment of the educational challenges the pandemic has brought in some countries [3]. Their analysis indicates that a series of actions are necessary to provide an appropriate educational response to the emergency phase. These actions include online learning, as it provides the greatest degree of versatility and opportunity for interaction and collaboration among students to foster mutual learning and well-being.

During the emergency phase, various research institutions developed guidelines and described online learning strategies implemented during the COVID-19 outbreak [4]. In these contributions, the different authors have suggested using synchronous and asynchronous learning and adopting various technological tools and teaching methodologies to help students build their own learning. Therefore, to ensure effective online education, flexible and active learning has been required. However, to deal with the emergency successfully, it is not only necessary to change the form of established learning and teaching, but also to redesign the online teaching model [5].

The prolongation of the crisis in the school year 2020/21 has required the establishment of new educational scenarios, teaching routines, and flexible settings aimed at the active participation of students, even though they do not follow standard timetabling nor occupy normal classrooms. The Italian school context after the first emergency phase in March–June 2020 has been regulated by DPCM 25 October 2020 as follows: during the school year 2020/21 in primary school and middle school, in-person teaching returned, while for secondary school, distance education was maintained for 75% of school time; as a result, difficulties arose concerning students with medical problems, who could not return in person due to the risks related to COVID-19 infection.

In the Italian context, Homebound Education (HE) service guarantees students the right to study, as stipulated in the Italian Constitution, even when they cannot attend school in person due to their health conditions, and before COVID-19 spread, one or more teachers went to student's homes to give lessons.

However, during the pandemic phase, the management of students with medical conditions or discharged after hospitalization was complicated by their physical vulnerability: the risk of contagion precluded their return to classroom or the teachers' presence at a student's home.

In light of the above, the solution of a hybrid learning environment [6] was used to support the regular school attendance of two students who could not return to their classrooms.

This paper reports the results of the interviews conducted with the teachers of the schools attended by the two students aimed at exploring their experience with a hybrid learning environment.

The next section will outline the functioning of the HE and the theoretical framework of the Hybrid Inclusive Classroom (HIC) developed in the context of TRIS (Tecnologie di Rete e Inclusione Socio-educativa; Network Technologies and Socio-educational Inclusion) [6] project, already tested before the pandemic.

## 2. The Homebound Education Service and Related Application Limits to Chronic Disease Cases

In Italy, HE started to be widely applied in early 2000s, as a result of increased awareness about the educational pathway of sick pupils, especially for those with serious illnesses who cannot attend their schools regularly.

HE is an extraordinary and temporary intervention, preparatory to the pupil's return to regular class [7]. Its purpose is to enable homebound students (HBSs) to counteract detachment from pre-disease school daily life and to maintain their social and friendship relations within the school environment.

In the Italian context to access HE, specific administrative procedures must be respected. Furthermore, HE can be carried out in three different scenarios:

- The children or young people receive HE in their own homes;
- The children or young people receive HE in other residential communities;
- The children or young people receive HE at the hospital in case a school service is not present.

In the first case, the teachers, who carry out the HE, are the same teachers of the class of HBS. In the second and third cases, the HBS will relate with strangers.

According to the service activation procedures, the student's school councils drafted an educational project, indicating the number of teachers involved, the subject areas to be prioritized and the hours of lessons scheduled. Overall, the lesson time is approximately 4/5 h per week for primary schools and 6/7 h per week for middle and secondary schools.

Throughout the time of HE service implementation, several critical aspects have been highlighted, in particular those related to the lack or reduction of contacts and relationships between the HBS and his/her mainstream class [8], which can be considered a risk factor for increasing the student's psycho-physical vulnerability. Through HE, academic instruction is granted, although it is perceived as a poor substitute for real school by

students themselves [9,10], who often experience feelings of isolation and loneliness [11–13], exacerbated by the impossibility of attending school trips and extracurricular activities [14].

Maintaining social and educational ties with the mainstream school provides to BSHs a sense of normalcy, that not everything has been wiped out by an invalidating illness. Moreover, theories on the positive effects of social support [15] state that the relationships with classmates mitigate the negative experiences connected to diseases and increase their sense of control and help them to face treatment better [9]. Interpersonal relationships are functional for healthy and harmonious growth, and they have a fundamental influence on the development of our minds [16].

In a review concerning studies on the special educational needs of chronically ill adolescents [17], it emerges how school, through the opportunity of social networking, could fulfil the student's need for normality and conformity' satisfied with the development of friendships and a sense of belonging to a group. The pupil's sense of belonging to a network and relationships with peers can develop social skills that support school performance and ensure effectiveness in adult relationships [18].

In a systematic meta-review [19], concerning school experiences of chronic disease students (affected by asthma, cancer, cystic fibrosis, gastrointestinal disease and heart disease), some of the studies analyzed, revealed that the academic achievements of these students are lower than the performance of their peers [20,21]. Furthermore, school attendance is lower and relationships with their peers and teachers are poorer compared with students without chronic illness. In some cases, because of their appearance, students with chronic diseases were bullied by classmates [19].

Children with different and chronic illnesses may experience a lack of control over their school performance, less interest and involvement, as well as feeling socially isolated [22].

This evidence suggests that children and adolescents with chronically disabling health conditions are more likely to experience a sense of isolation and the loss of the important social and peer contact dimension that school traditionally provides. Lum et al. [19] point out that the engagement of the student with chronic illness with his/her mainstream class is necessary for his/her academic and social needs; in this line, a HIC setting could be a suitable response to the issues raised and a promising approach to connect distant students with chronic illness [23].

## 3. The Theoretical Framework

### 3.1. From Hybrid Virtual Classroom

Over the past decade, the increase in the uses of mobile devices, supported by the spreading of connectivity infrastructures, has enabled students, from potentially anywhere, to include these technologies in their learning paths, leading to a redefinition of the spatial boundaries of learning environments (hence the concept of "ubiquitous learning") [24].

These new environments are defined as hybrid spaces and originated from the fusion of different physical spaces, real or virtual, sometimes integrated or overlapping. Hybrid spaces can enhance learning opportunities, integrating formal settings with informal setting experiences [25].

An important aspect that distinguishes a "hybrid space" is a change in the concept of proximity: the use of connected devices creates the conditions of social actions, putting in contact geographically distant people (or who just cannot meet in person), ensuring them the possibility of interacting [26].

We can define *hybrid* as the space that is created out of the interrelation of the online and offline environments in which people set their working/studying/encountering moments. This allows us to assume different educational, participatory and inclusive scenarios that overcome the dichotomy between in-person and distance teaching [6].

Virtual environments can be used to hybridize the different physical spaces where actors are located, even supporting distance learning activities (hybridization of teaching spaces with remote students).

The use of digital resources plays a participatory and social role, fostering the creation of educational pathways that ensure that everyone could be actively involved, each with his or her own needs.

A recent analysis of the literature [27] highlights how learning in a synchronous hybrid space can represent a more flexible and engaging environment than those of full physical presence.

Authors emphasize the benefits from an organizational and pedagogical point of view and the changes required to organize and manage a hybrid learning setting. Some contributions analyzed in the review highlight how a hybrid setting can support inclusion and equity in the educational context [28,29] and reduce teacher's workload by preventing the same course from having to be replicated multiple times [30,31], and how collaboration and connection between face-to-face and HBSs create richer learning experiences [28,30].

Implementing a hybrid classroom requires, in addition to technological skills, a change in the methodological approach [28,29] that can effectively integrate technologies [32–34] to facilitate the social presence of remote students [35].

### 3.2. To Hybrid Inclusive Classroom

The hybrid classroom can offer a solution for HE services, solving many of the limitations highlighted in the literature, especially referring to social dimension issues and behavioral and cognitive engagement of HBSs [36].

The indications related to the hybrid space have stimulated the development of an ecosystem model [37], centered on the concept of an HIC to promote the social and educational inclusion of HBSs. The concept of HIC was established in the TRIS project [6] that has been carried out thanks to a three-year framework agreement signed between the Italian Ministry of University and Research (MIUR), the National Research Council (CNR), and the Telecom Italia Foundation (social outreach arm of a major telecoms provider). The project lasted three years and involved four students affected by chronic diseases without cognitive impairments, their teachers (47) and their classmates (over 80 students).

Students involved in the TRIS research project were selected because they would not benefit from HE in any way (due to the nature of their disease) and there was no option for them to return in person.

Specifically, these students were permanently prevented from regular classroom attendance, posing significant critical issues in the use of the traditional HE service and their educational and social inclusion. In addition, in the case of three students, teachers were unable to have in-person interactions due to their physical vulnerability.

In this context, a new space for interaction has been designed for daily, synchronous and continuous participation of the HBS to class activities and school life in general, attending lessons, participating in classroom discussions and posing questions, carrying out tasks and checks with corrections in real time, and participating in activities workshops and group activities.

The hybridization supported by the technological and digital setting therefore has developed a new physical/virtual and interactive space that arises from the fusion of the traditional school environment and the HBS's home. In Figure 1, there is a graphic representation of the model.

The school space that was built has therefore been supported by online resources that make it possible to interact, share and collaborate among students. The use of technology, however, was only a part in a more radical transformation that concerns the very organization of spaces, and, above all, the didactic strategies adopted. The approaches more oriented to the collaborative and active dimension of students, in fact, were important enabling factors for the inclusion of students who could not attend the lessons in person [38].

From direct observation in the classroom and the qualitative/quantitative analysis of data collected emerged a model with four dimensions.

- Contextual
- Organizational

- Technological
- Methodological

For each of these dimensions, a set of actions, factors and/or hints have been identified that should be taken into consideration when implementing HIC.

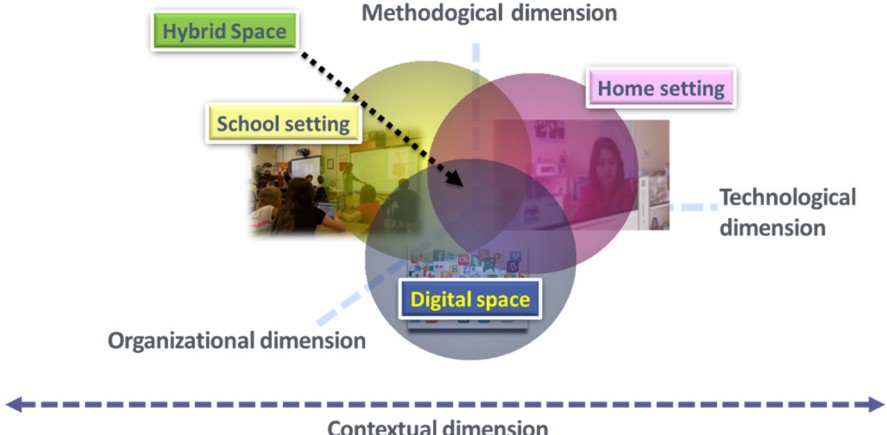

**Figure 1.** Hybrid educational model.

### 3.2.1. The Contextual Dimension

The contextual dimension is related to the wider ecosystem involved to understand, support, and improve the complex relationships between teachers, parents and families. The activation of HIC requires teachers to reconfigure relationships aimed at creating and fostering a participatory school and educational context.

### 3.2.2. The Organizational Dimension

The organizational dimension concerns the organization of the classroom and home spaces in a manner suitable for teaching in a HIC, entails redefining or creating new routines specifically to fit the new context and organization of teaching times.

Handling HIC requires the adoption of teaching solutions that are based on the active and collaborative participation of each student (whether present or distant) and the consequent pursuit of organizational arrangements capable of integrating technology by making it as "transparent" as possible.

### 3.2.3. The Technological Dimension

The technological dimension concerns the study of sustainable technological infrastructure setups (class and home side) and the identification of mobile technologies and cloud resources capable of satisfying three typical functionalities: interpersonal communication, sharing of resources and co-construction of artifacts.

### 3.2.4. The Methodological Dimension

The methodological dimension concerns the didactic-pedagogical approach adopted to foster active involvement of the HBS in learning activities and social interaction with their classmates. Having a HBS as a class member requires teachers to manage classroom activities in a manner that differs from traditional teaching situations. Experience with the learning environment adopted during TRIS project has generated some ideas and possible indications for ways of supporting distance teaching solutions. In Table 1, we summarize the different actions and the suggestions related to each dimension that can support the teachers to implement HIC with HBS.

**Table 1.** The dimensions related of HIC.

| **Contextual Dimension** |
|:---:|
| Need for collegiality in the management of technology<br>Need for collegiality in teaching |
| Educational partnership between school and family |
| Creating a climate of trust with the family<br>Need for distance mediation by an adult<br>Management of the home setting<br>Management of family interference |
| Relationships among students |
| Maintaining emotional connections (BOND) |
| **Organizational Dimension** |
| Organization of physical and virtual spaces |
| Organize physical spaces<br>Organize learning online environments |
| Relationship routines |
| Relationships routines to facilitate the teacher's work<br>Establish and shift the virtual desk mate<br>Relationships routines to facilitate a feeling of friendship |
| Behavioral rules |
| Application of pre-existing rules for the homebound student and class<br>Develop new rules for the homebound student |
| **Technological Dimension** |
| Factors hindering use technologies |
| Inadequate sense of self-efficacy in the use of technologies<br>Difficulty in managing the technological setting<br>Resistance to technology integration |
| Prerequisites for the use of technologies in the classroom |
| Need for basic skills in the use of ICT<br>Need for time to experiment with teaching technologies |
| Technology Affordances |
| Affordance/collaboration<br>Affordance/access to resources |
| Positive factors in using technologies |
| Technology as a tool for didactic innovation<br>Technology as an inclusion tool |
| **Methodological Dimension** |
| Planning educational activities |
| Need to plan educational activities<br>Alternate between synchronous and asynchronous activity |
| Teachers' Educational Approach |
| Perception of change in one's professional practice<br>Overcoming the lecture-style teaching method |
| Learning and evaluation strategies |
| Active Learning<br>Peer tutoring<br>Pair learning<br>Evaluation strategies equal to the rest of the class |

## 4. Materials and Methods

### 4.1. The Present Study

The present study was carried out in the context of HE projects, run in the school year 2020/21, for two students suffering from acute or chronic diseases (cancer and cystic fibrosis).

Due to the prolongation of the pandemic emergency and according to the dispositions of the Ministry of Education regarding the integrated digital didactic (DDI) for students with serious illness or who are immunocompromised (O.M. n 134/2020), the individualized educational path provided for HBS should be integrated with distance activities with the classes. This context has opened an opportunity to innovate traditional HE toward HIC [6]; in fact, HIC environments could support the traditional HE service in Italy and enhance the social and educational inclusion of students forced to stay physically outside of their classrooms.

Starting from these premises, the collaboration with the schools was initiated within the framework of the CLIPSO project (hybrid classes for SiHo; https://www.progetto-clipso.it/) (accessed on 6 August 2022), which aims to overcome the isolation of hospitalized children and to support their inclusion in the curricular activities of their mainstream schools. Firstly, in the context of action research, teachers have been supported and trained on the HIC model [39]; then, they were assisted as needed in carrying out learning activities or managing the technology setting during the school year.

The main aim of the present qualitative study is to explore teachers' experience with the new learning environment in order to get preliminary indications for its use in HE.

### 4.2. Methods

#### 4.2.1. Participants

The study involved 22 teachers belonging to two lower secondary school councils of the HBSs (15 female and 7 male). A total of 17 teachers took part in the final interview sessions.

#### 4.2.2. Procedure

The study was carried out using a qualitative methodology, specifically a group interview [40]. The text of the interview was developed by the members of the research team. The text of the interview is available in Appendix A.

Two separated interviews sessions were conducted online and carried out during the last week of May 2021. The different sessions were video recorded. Before any interviewing activities started, participants gave informed consent to the recording, processing, and use of the material collected. All the interview sessions were transcribed verbatim by a member of the research team.

### 4.3. Analysis of the Interviews

Given the nature of the research goals, related to understand the impact of the HIC model to support HE, the analysis of the interviews was carried out using a deductive top-down approach that was based on the model's categories. Therefore, two researchers coded the contents of the interviews according to the codebook (a set of codes and related definitions, overarching themes and dimensions) developed in the context of TRIS project, characterizing the HIC model [41].

At the beginning of the analysis process, the coders (two psychologists who conducted the interviews) familiarized themselves with interview transcripts through multiple readings. Then, they coded the texts of the interviews separately, according to the codebook and also identified new emerging codes. Subsequently, they compared their code category sets and interpretive differences were resolved through discussion and negotiation. This ensured that coding consistency was achieved through a two-step process: individual parallel coding and subsequent negotiation of differences in the generated codes.

Then, a new codebook was developed through the combination of the early codebook with the new emerged codes and themes.

Afterwards, the two researchers coded the interviews in light of the codebook developed, and after the coding process was completed, a joint review was conducted to resolve any discrepancies.

## 5. Results

In the following, we present the results of the analysis organized under the different dimensions of HIC. An overview of final themes and details are presented in Table 2, including examples of main codes, their description and example quotes.

**Table 2.** Factors and actions related to different dimensions of HIC.

| Contextual Dimension | | |
|---|---|---|
| **Theme Name** | **Theme Description** | **Example Quotes** |
| **Relationships between school and family** | Teacher recognizes the need of a cooperative climate with the family. Teacher also recognizes some critical issues | *"I know for sure that, if all the homework is well done, parents helped their son in doing the work."* *"The family atmosphere is very respectful, relaxed, so from student's parents there wasn't any interference"* |
| **Relationships among and with the students** | Teacher recognizes his/her difficulties and those of his/her students in interacting with HBS. Teacher also identifies a series of behaviors aimed at maintaining contact with HBS | *"Some classmates were insensitive toward X, who thought she was privileged"* *"I was afraid that I would not be able to focus myself on X as much as I wanted to* |
| **HBS characteristics** | Teacher indicates that HBS personality or his/her health condition may interfere with the HIC | *"I knew his/her personality, strictness and self-control would help him/her!"* *"He/she's not outgoing, that's helped him/her in studying but not in relationships."* |
| **Collegiality and the role of teachers** | Teacher affirms the need of colleagues' cooperation, especially with those involved in personalized teaching activities | *"I'm grateful for my colleague's support. Without her we would had struggle more."* |
| **Organizational Dimension** | | |
| **Setting organization** | Teacher thinks that managing the HIC setting requires changes in the classroom traditional setting | *"The class group, in which X was part, was connected in videocall with X (using tablet). For this reason, group with X worked in the hallway, to avoid that the rest of the class cause audio difficulties during videocall."* |
| **Relationship routines** | Teacher considers routines as necessary for class relationships, still admitting some critical points | *"I consider necessary that a classmate helps HBS . . . however, in some cases, I noticed that this classmate struggled to keep up with lesson because of it, so it's a role that we teachers have to be careful to assign."* |
| **Technological routines** | Teacher explains the procedures and routines for starting videocall and handling it, involving HBS and his/her class during daily lessons | *"We included the use of the tablet by the classmate on duty, so during lessons X was connected with teacher and classmate both."* |

**Table 2.** *Cont.*

| Contextual Dimension | | |
|---|---|---|
| **Theme Name** | **Theme Description** | **Example Quotes** |
| **Technological Dimension** | | |
| **Use of different technological resources** | Teacher says HIC setting requires several technological resources, finalized to different purposes, such as communication, lesson material storage and so on. | *"Regarding the virtual environment (HIC), I have to say that the presence of X gave me the reason to using the tech as much as possible. I was not used to it, but tech is an important resource, even more important if there is someone at home who needs to have the lesson material. So, it was an opportunity for me to regularize and continually update our virtual environment."* |
| **Technology as a tool of inclusion** | Teacher recognizes that technologies are an important tool for fostering inclusion | *"I have two students with special needs in my classes, and everything we had already used with X was useful for those with other needs as well."* |
| **Positive factors in using technologies** | Teacher claims that prior technologies experiences, before this research, have fostered HIC management. In particular, teacher is mentioning the experience of remote teaching during lockdown | *"It wasn't out of the blue, with this experience I continued to train and do these things, because personally, when I started distance learning last year, I didn't know so many things that I learned gradually."* |
| **Critical factors in using technologies** | Teacher observes several difficulties related to the setting, such as noise, connection difficulties and so on | *"There were audio and connection problems, that sometimes make it difficult to connect with HBS."* |
| **Methodological Dimension** | | |
| **Planning educational activities** | Teacher considers it necessary to plan teaching activities in HIC setting | *"So, it wasn't easy for us (teachers). I mean, I had to prepare materials, I learned how to schedule everything, the emails, the activities to do, and I was afraid of forgetting something."* |
| **Teachers' Educational Approach** | Teacher describes teaching experience with the HBS as a professional gratification and a matter of pride for institution. | *"I think this pilot experience is meaningful for the territory, not only for our school but for all Italian schools dealing with home schooling."* |
| **Learning and evaluation strategies** | Teacher claims to have favored active participation and group work strategies to foster socialization as well, and to have adopted the same strategies for both HBS and classmates | *"I adopted the 'small group' setting to get him/her in touch with some of his/her classmates, to do homework together or have some not-so-formal moments . . . "* *"In my opinion, from learning perspective, it seems to me that he/she was able to keep up with what we were doing in class, he/she was always careful, he/she always had very good results."* |
| **Hybrid Class Experience** | | |
| **Increased workload** | Teacher affirms that teaching strategies for the HIC setting result in an increased workload | *"I struggled here, because in fact the use of different tools, different way of teaching, the planning . . . in short, I had to work hard."* |
| **HE importance** | Teacher says that traditional HE lessons (1:1 at student's home) were effective | *"During individual lessons we did a lot more than what I do in class, because he/she understand better when we interact directly, so it was absolutely positive."* |

**Table 2.** *Cont.*

| Contextual Dimension | | |
|---|---|---|
| **Theme Name** | **Theme Description** | **Example Quotes** |
| **Use of HIC for absent HBS's classmates** | Teacher says that, due to the HIC experience, he/she found easier to use remote learning setting with absent students as well | *"No, I just wanted to say that this experience makes me . . . other times with other students who were, because of long absence or lockdown, forced to stay at home, made the whole thing very normal, you know, even students at a distance were participating a lot to the lessons."* |
| **Perceived equality when all class were in remote setting (because of lockdown)** | Teacher reports that remote setting due to lockdown made the HBS feel on the same level as others and increased his/her perception of equality | *"The other good experience for X was distance learning for the whole class. At that point he/she really felt equal, I had already said that: he/she really opened up whit me, telling me that this context finally put him/her on the same level as the others."* |
| **Active involvement of all students when class was in remote setting** | Teacher reports that the class actively participated in remote lessons during lockdown | *"Of course, in that emergency phase, for me it was all easier, everyone is equal, and so everyone participates, respects others . . . I find them as a beautiful class even during distance learning, I mean, despite of everything, there is broad participation."* |

### 5.1. The Contextual Dimension

Regarding the relationship with the family, the teachers report the presence of a bond of trust and mutual support:

*"I consider the student's mother to be very precise. She always made an effort to give us the requested information and materials in real time: handing over the tests done by N. at school, picking up homework, she never backed out he gave so much . . . /She always answered to any request, to any message promptly."*

In addition, teachers have a perception of possible HBS's family interference in the educational setting:

*"As might be expected, we can check if all homework was done by the homebound student, or someone made them for her. But at home she has more freedom and that's right"*.

Regarding their relationship with the HBS, teachers report that they are concerned about her/his loneliness:

*"Make him/her feel important, despite the distance, is an essential part of the class purpose. I think it's the fundamental thing . . . "*, *"because distance could make the homebound student think: 'They are in classroom, I'm at home, we are not the same'"*.

Teachers identify a set of actions required to facilitate the social inclusion of HBS:

*"We pushed him/her, we told him/her 'well done' in all ways, from this perspective I think this support helped him/her feel like as he/she is part of the class"*.

Additionally, during the interview, teachers' emotional involvement with the student's situation emerged: *"In my mind, every time I think of him/her or see him/her for an interview ( . . . ) or when we directly get in touch I can't help being touched, because he/she touches my heart"*.

Furthermore, the teachers recognize the need to normalize the new school setting:

*" . . . every now and then I say something to him/her, however, I tried to not draw much attention to him/her during my lesson. So far that if I didn't say hello to him/her . . . because I also have the same problem as my colleague, who says "I didn't say hello to*

*him/her". I don't even go back to say hello to him/her, because I don't say hello to them all, I say "hello guys", I don't say "hello A, hello B, hi C . . . ".*

However, some students' personality traits have an impact on social inclusion in the school environment, such as introversion:

*"Maybe somehow we found a way to help him/her in him/her relations with peers. He/she is not an outgoing person, particularly extroverted in short, so what helped him/her in teaching (introversion) maybe it was an obstacle in create bonds with classmates. I would say maybe another person could have, even at a distance, created more relationships."*

However, teachers also point out some attitudes and behaviors of HBSs that, in addition to being positively evaluated, are considered as aspects that promote their inclusion in the classroom setting: *" . . . he/she was very accurate, precise and independent, and I told him/her several times that many things are taken for granted by adults, but they're not. I see adults struggling to do a whole series of operations that he/she easily learned to do, like downloading the file, printing it, working on it . . . ".*

In terms of the relational dimension between classmates and the HBS, critical issues and conflicts are reported, although some of these fall within typical peer relationship dynamics:

*"I mean, there were dynamics that also involved . . . but they were class dynamics, even if he/she was from home, even if there was an unpleasant episode . . . I don't know how to say it . . . well, in some ways it was really a class dynamic so . . . even though of conflict, that seemed to me a positive experience as well".*

Additional difficulties include situations in which the HBS was perceived as privileged by classmates: *"there was some tension between A and her classmates who considered her a bit privileged".*

The mentioned difficulties have not prevented students from building even deep affective bonds, including the love dimension: *"there was even a distance romance that was born and grew through the tablet, also with some outside in-person contact, probably, but for the most part experienced using the tablet".*

### 5.2. The Organizational Dimension

As far as the organizational dimension, teachers identified some routines to be able to better manage the communication with the HBS and the classroom:

*"Then I used to put earphones on, and I could hear him better even if there was noise'.*

Teachers report that classmates had a key role in supporting both themselves and the HBS and consider the virtual desk- mate role to be effective:

*"Students have acquired an impressive routine, in the sense that they are completely autonomous in handling technology".*

*"The tablet used by the student was not connected with the teacher's Meet; so, I didn't really understand technically how this thing was, maybe C could explain it better, I don't know, but he/she was connected with the teacher and with a companion".*

Additionally, the new setting has been positively perceived by the whole class:

*"( . . . ) the nice and positive thing is that his/her classmates have been always very happy to help him/her from remotely, of course the situations in which this occurred were not many, but it was a good experience for P and for the boys in the class . . . ".*

Teachers also mention the need of actions in order to manage a virtual space aimed at sharing teaching materials with the HBS:

*"As far as the virtual environment is concerned, I must say that the presence of A for me was the key to accustoming myself to using it as much as possible. I wasn't used to it and it is an important resource, even more important if there is someone at home who needs to have the material at hand. It was therefore an opportunity to regularise and continually update the virtual environment section".*

Several critical issues in the simultaneous management of the two settings (face-to-face and virtual space) are reported: " *... sometimes amidst the noise of the class you can't talk so well, I don't understand it*". The two settings even slowed down the course of teaching activities: "*Having to explain one thing a hundred times to the student at home who did not understand when the class was at another point was very tiring for me*".

However, some teachers state that the noise and confusion that the HBS heard in the classroom could foster a greater sense of inclusion: "*I think this chaos makes him/her somewhat happy, because anyway even hearing us saying "you shut up", the kid making the joke ... this we say is part of her everyday life*".

### 5.3. The Technological Dimension

Teachers state the usefulness of some technologies for managing the HIC setting: "*it helped us that we used Classroom ... if I gave to the others in presence the printed sheets for a test, I would send to him/her (the digital version)*".

The coexistence of different technologies that enhance the inclusion of HBS was crucial, according to teachers' thoughts:

> "*He/she did also asking questions in the chat room, so his/her classmates kept quiet when he/she would speak*".

The previous use of technologies to activate distance learning during the pandemic emergency is reported as facilitative in the implementation of HIC: "*So, compared to the beginning of the year, I didn't have any particular concerns related to the technical aspect because we had our own experiences, last year we had done an entire four-month in a distance setting*".

However, teachers consider as fundamental regular technologies courses for them, to achieve a minimum range of autonomy. In addition, they argue that the technological setting of HIC has forced them to deal simultaneously with a plurality of technologies: "*So, first of all, for me it was exciting. I couldn't tell that I was familiar with tech in the beginning -fortunately I had already used Classroom last year, so I had some early knowledge. But ... for example, I had never used the interactive whiteboard, I had never used the Jamboard, I had never used tablets, that is several things that, with this experience, I tried to learn. Maybe not in a perfect way, however just enough*".

Teachers also state how students quickly and successfully gained digital skills: "*Because they are so good with technology ... they were able to figure out how it worked. Even N (homebound student) helped me figure it out at a glance ... a critical point where I couldn't proceed. I didn't understand I had to enroll in Padlet in order to access it. So, it was N who said to me, "look maybe you have to go and sign up!"*".

Teachers emphasize the importance of the tablet in fostering socialization and relationships with classmates and the HBS: "*I have to say that this thing (the tablet) in my opinion has been vital for the relationship. I leave aside the didactic part, which in my opinion is a bit more difficult ... well even relationships are an aspect that at a distance is complicated, however in my opinion from that point of view really the use of the tablet has been astonishing*".

Lastly, they point out the difficulties related to connection and audio, as well as the misuse of technologies by classmates during lessons: "*We had a few episodes during the year ... technology using that were a little, perhaps, not consonant, so it happens that someone watched contents not related to the lesson for his own business*".

### 5.4. The Methodological Dimension

Regarding the methodological dimension, teachers state the need for designing their learning activities:

> "*I would prepare, I learned how to plan. Then I would write all the emails with everything the student at home had to do ... I was afraid I would forget everything*".

They consider it necessary to personalize the learning pathway in relation to the needs associated with the illness:

> "*We have to keep in mind the person ... we knew she was ill and might have specific needs*".

As far as teaching methods, the collaborative approach was preferred to the extent that teachers emphasize the relational dimension that naturally supports mutual understanding between students:

"... *the fact also of convening small groups and working in a certain way with them helps a lot from this point of view, I tried as much as possible ... yes because I think the art curriculum is fundamental, but it is also fundamental to grow together ...* ".

"... *at the beginning, it seemed that mainly those who already knew him were involved, those who supported him because they knew a bit of his history and so they made themselves immediately available, and then, perhaps, one or two classmates who didn't know him had joined in, instead ...* ", even though in some moments teachers also point out some difficulties: "*I know that that group didn't always work very hard*".

Teachers in technical subjects complained about the workload due to remote teaching:

"*Because my subject in remote education is a disaster ... because teaching drawing at a distance is really problematic and I realized this*".

As far as HBS's evaluation, teachers report that they relied on the same summative strategies used with the classmates:

"... *the other day, just yesterday, we gave her a round of applause because even at the level of learning, I don't know how she did it-I do one hour and a half hours of French, the last two hours until a quarter past two-she's always there, she's always done everything, we did a listening comprehension test that objectively is impossible to copy, difficult, maybe with the audio not good and she showed that even at the level of learning she held perfectly ...* ".

Teachers also claim to have carried out formative evaluation aimed at discussing the difficulties faced by the HBS: "*You know, it happened that some tests didn't go so well, I gave him/her an individual appointment in the afternoon, she naturally came without batting an eyelid, we talked to each other, we talked about the problems of the tests*".

Overall, teachers argue that the HIC approach allowed the HBS to reach a learning level similar or comparable to that of the classmates: "*It seems to be a miracle that this boy/girl has kept up with the others without ever having set foot in school, so I don't even know what to say*".

*5.5. Hybrid Inclusive Classroom Experience*

Teachers recognize the role of the HIC in promoting the inclusion of the HBS by perceiving him/her as being there in person: "*I must say that for me it is always almost a miracle that there is the possibility for a sick person to be able to follow and be among his/her classmates, of course with all the limitations*".

They underline the need for integrating the traditional HE with the HIC: "*With home-bound education it was possible to supplement what was missing, so I think it was a positive experience for him/her*".

Furthermore, teachers state that the HIC can be extended to other students not attending: "*This has made it possible for students who are absent for a long time or are in quarantine to participate at a distance*".

According to the teachers, the presence of the HBS has added value to their personal and work experience: "*A moment that humanly served us, him/her and the class and transported us, as it were, to a more human, warmer level than everything that technology has made us do this year, which however has made us all take huge steps in one way or another. Here, in my opinion, it was like a brighter dowel within this making and making do and finding strategies for everyone. So instead of being a burden in some ways it has turned into an added value*".

## 6. Discussion

As regards the context dimension, the teachers underline the importance of the relational dimension of all the actors involved: in order to implement the new school setting, it is necessary to motivate the students' families to collaborate, making them more

aware of their active role in their children's learning path through sharing an educational agreement [41]. The implementation of HIC also requires a bond of trust between schools and family and a close and frequent collaboration, considering that online learning activities during school hours impact on "new" and different spaces (i.e., the student's home) [42]. According to the results, however, the teachers' perception of their relationship with parents is positive. That differs from the data of a further national survey on HE [8] in which the relationship with parents is perceived as a stress factor: in fact, the involvement and the request for psychological support of families can be a stressor factor in the teachers' professional practice [43].

Despite the potential emotional burden associated with dealing with an ill student, the teachers involved in this research were particularly responsive and the presence of the HBS prompted them to pay special attention to the relational dimension with students. In fact, the results have shown that the teachers implemented a variety of actions in order to facilitate the HBS participation in the social life of their class, with structured activities such as small groups and informal activities, thus facing several critical aspects, as reported in previous studies, related to the loneliness and lack of social support that HBSs usually complain of [18,19].

The analyses also reveal a rather interesting result related to the role of the HBS's temperamental characteristics on his/her inclusion, a finding that had not emerged in previous researches. Teachers, in fact, emphasize how important it is to know the student's features in order to support them, since, for example, aspects related to shyness could be inhibiting the active participation in the hybrid class context. An additional important issue to consider is related to the health status of the student: in some cases [8,25], it was highlighted that students did not want to show themselves on video because their physical state had undergone changes (i.e., hair loss) during medical treatments. Acquiring as much information as possible about the student's psychophysical state is therefore necessary to avoid the educational setting from being a source of further stress and suffering.

From an organizational perspective, the management of HIC can be facilitated through the implementation of some technological and social routines.

In addition, teachers consider a variety of organizational changes related to the setting to be necessary in order to make it more inclusive for distance students, as several authors have already pointed out [27]. The students' support in managing the technological and relational setting is also considered by the teachers as an added value; it is, in fact, a school environment certainly enriched by new tools that require flexible adaptation and additional workload, in some cases perceived as overloading by teachers [6].

Furthermore, teachers report as necessary a whole series of organizational changes related to the setting to make it more inclusive, for those at a distance, as already highlighted by several authors [27].

Regarding the technological dimension, teachers recognize that the use of technologies in the first phase of the COVID-19 related emergency was useful in order to force them to familiarize themselves with some technological resources never before tested [44].

Certainly, the pandemic largely reduced some of the hindering factors, such as the low sense of self-efficacy in using technologies and their poor integration into daily routines aspects that we can consider as "internal factors" [41,45–48].

In the context of the hybrid learning environment investigated in the present study, technologies are perceived as inclusive artifacts: for example, the tablet managed by the peer in the classroom to communicate with the remote mate can activate or support a process of social and educational inclusion.

A change in teachers' perspectives on the use of educational technology can then be fostered through their personal experience with technological solutions that can facilitate the inclusion of students unable to attend school normally, which can also potentially be used with other students with special educational needs [49].

With regard to the methodological dimension, teachers have declared the importance of activating a teaching environment based on the use of cooperative strategies in order to

promote learning and support the relationship between students considered as a necessary condition for fostering a sense of a sense class membership [17,18,50]. These findings underscore a difference between a traditional HE approach and the HIC. In fact, in the HE setting, the relationship with peers is almost always lacking and is usually difficult to establish because the service involves the a teacher's presence during out-of-school hours. The HIC setting overcomes this limitation, although moments of one-on-one interaction between teachers and the HBS are considered necessary to further support him/her in case of difficulties. The results show that students achieved the same levels of learning as their peers unlike what has been reported in other studies [20,21]. In addition, the presence of a teacher as a coordinator is important [43], as well as teamwork among teachers to share a joint project.

Finally, the adoption of the HIC seems to have facilitated the transition of the whole class from face-to-face to distance teaching during some moments of pandemic exacerbation. These results indicate how the daily use of technological resources can foster the integration of educational technologies into teachers' practice and support real processes of equity and inclusion.

## 7. Conclusions

The HIC marks a break with the school of the past. We are facing a paradigm shift that requires a new organization of the school environment, necessitating new instances and resources, but above all a cultural attitude toward innovative educational processes. The HIC requires the involvement of all actors; the teacher alone cannot manage the complexity of the setting. The articulated use of different technological resources indicates how school environments must be rethought based on new needs.

New dynamics and new roles distributed among all actors emerge.

It would be appropriate to enhance the roles of students as the tasks required of them promote those skills related to active citizenship (ref.).

The prolonged emergency related to COVID-19 showed that a physical return to the classroom for many students was not possible as they were at risk due to their vulnerability related to their health status. For these students, it was possible to activate HE, which involves teacher turnover in the student's home during extracurricular hours. As already pointed out while HE enables the right to study, the disruption of social ties with peers, sometimes prolonged in time, risks making students with chronic conditions even more vulnerable.

Analysis of the results shows how classical HE, through the adoption of HIC, can effectively respond to a whole range of critical issues that have emerged over the years: the difficulty of teachers in managing intimate interaction at the student's home, the lack of interaction between HBS and the home class, and the limitation of disciplines that a home student can take.

The findings reinforce the framework of the HIC, which with the indications that also emerged from this research enrich the actions and suggestions previously identified.

It would be desirable to move from the adoption of the HIC in an emergency context to an increasingly ecological use of an instructional approach that relates the different dimensions identified.

This principal limitation of the study is the circumscribed number of cases analyzed.

Further research would be desirable to understand the validity of the model and its enrichment beyond emergence issues.

**Author Contributions:** Conceptualization, V.B., G.P.C., F.M.D., E.D.M. and C.F.; methodology, V.B., G.P.C., F.M.D., E.D.M. and C.F.; formal analysis, V.B., F.M.D., E.D.M. and C.F.; investigation, V.B., G.P.C., F.M.D., E.D.M. and C.F.; writing—original draft preparation, V.B.; writing—review and editing, F.M.D., E.D.M.; supervision, C.F.; funding acquisition, V.B. All authors have read and agreed to the published version of the manuscript.

**Funding:** The CLIPSO research project was funded by Fondazione Compagnia di San Paolo.

**Institutional Review Board Statement:** The reported research project activities involved external subjects and have been reviewed and approved by the Commission for Research Ethics and Bioethics of the Italian National Research Council (CNR). Ethic Committee Name: Cinzia Caporale. Approval Code: 0043048/2019. Approval Date: 14/06/2019.

**Informed Consent Statement:** Informed consent was obtained from all subjects involved in the study.

**Data Availability Statement:** Data available on request due to restrictions, e.g., privacy or ethical. The data presented in this study are available on request from the corresponding author. The data are not publicly available due to privacy issues.

**Acknowledgments:** The authors wish to thank all the teachers and principals who participated in the study. We also wish to thank grad student Alessandro DeSanctis for his contribution in data analysis.

**Conflicts of Interest:** The authors declare no conflict of interest.

**Appendix A**

Preliminary questions for class teachers (irrespective of the systematic start of distance learning activities)

- Before your current experience with the HBS . . . were you familiar with the School in Hospital service?
- According to your experience, what strategies can be useful to maintain educational continuity while a student is hospitalised or absent due to illness?
- According to your experience, what strategies can be useful to maintain social and relational contacts between the hospitalised student and his/her classmates?
- What impact did the student's hospitalisation have on the class? Did you have to manage critical issues? If so, how were they handled?
- How did things go with the HBS during the medical emergency and the remote learning during lockdown?
- Communication with School in Hospital teachers

For cases in which it was possible to start distance activities with hbs (interview with teachers on hic) management

- Was the class equipped with technological tools?
- Were you already using technologies?
- What impact did the introduction of technology have in the 'classroom' context?
- How did you manage to use technology? Were there any changes during the school year (e.g., did you get used to it)?
- Have class 'routines' for technology management been developed (e.g., someone who sets up the PC, launches programmes)?
- Were there any interference elements of this new classroom environment that you had to manage?
- Have you developed new and different organisational management strategies for the 'new classroom environment'?

Communication

- How did you manage your communication with the HBS and the class?
- How did you manage communication between the students and the HBS?
- Did you need to establish new rules in your interactions following the construction of this 'new class environment'?

Family

- How were the relations with the families involved (HBS and class)?

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
