# Peer review of "Enhancing Home Education in Italian Context: Teachers’ Perception of a Hybrid Inclusive Classroom"

_education, doi:10.3390/educsci12080563_

Round 1

Reviewer 1 Report

An interesting proposal. The theme is relevant and not much discussed. It is clearly written and well-structured, and clearly explained conceptually and methodologically. The discussion of the data is interesting and the conclusions show consistency in being supported by the evidence. I consider it to be accepted without amendments.

Some minor corrections: in L.23, where you write 'provides' you should write 'provide', and in L.327 where you write ‘action research’ you should write 'action-research’.

Author Response

  • An interesting proposal. The theme is relevant and not much discussed. It is clearly written and well-structured, and clearly explained conceptually and methodologically. The discussion of the data is interesting and the conclusions show consistency in being supported by the evidence. I consider it to be accepted without amendments.

Response:

 We thank the reviewer for the comments, and we hope this contribution will be really useful to improve the HE not only in our country.

  • Some minor corrections: in L.23, where you write 'provides' you should write 'provide', and in L.327 where you write ‘action research’ you should write 'action-research’.

Response:

We revised and corrected the errors.

Reviewer 2 Report

The paragraph structures in several places need some work. There are times when it abruptly changes. Much of this is due to a lot of one-sentence paragraphs that are often so short that it is unclear what the purpose of the paragraph is as there are no supporting statements. 

Overall research and results were good. One issue is that the author stated this was a mixed method using both quantitative and qualitative methods, but only qualitative data was presented. It was not clear if quantitative data was ever collected or analyzed. I would suggest that if qualitative data was not used, simply state that this is only using a qualitative study or state that this manuscript is only focusing on the qualitative data.

Author Response

  1. The paragraph structures in several places need some work. There are times when it abruptly changes. Much of this is due to a lot of one-sentence paragraphs that are often so short that it is unclear what the purpose of the paragraph is as there are no supporting statements.

Response:

Thank you so much for your comments that were highly insightful and enabled us to greatly improve the quality of our manuscript. You’ll find signed in yellow the paragraphs that we changed

  1. Overall research and results were good. One issue is that the author stated this was a mixed method using both quantitative and qualitative methods, but only qualitative data was presented. It was not clear if quantitative data was ever collected or analyzed. I would suggest that if qualitative data was not used, simply state that this is only using a qualitative study or state that this manuscript is only focusing on the qualitative data.

Response:

Thank you for your question. We used “ both quantitative and qualitative methods” in the previous research thanks to which we developed the model of the hybrid inclusive classroom and you find it at L. 218.
While in relation to the specific analysis reported in this paper, you find the description of qualitative approach adopted at L.281.